# Using patient biomarker time series to determine mortality risk in hospitalised COVID-19 patients: A comparative analysis across two New York hospitals

Ben Lambert[1,2]☯*, Isaac J. Stopard[3]☯, Amir Momeni-Boroujeni[4], Rachelle Mendoza[5], Alejandro Zuretti[6]

**1** Department of Computer Science, University of Oxford, Oxford, Oxfordshire, United Kingdom, **2** Department of Mathematics, College of Engineering, Mathematics and Physical Sciences, University of Exeter, Exeter, United Kingdom, **3** MRC Centre for Global Infectious Disease Analysis, School of Public Health, Faculty of Medicine, Imperial College London, London, United Kingdom, **4** Department of Pathology, Memorial Sloan Kettering Cancer Center, New York, NY, United States of America, **5** Department of Pathology, SUNY Downstate Health Sciences University, Brooklyn, NY, United States of America, **6** Department of Pathology, SUNY Downstate Health Sciences University and Maimonides Medical Center, Brooklyn, NY, United States of America

☯ These authors contributed equally to this work.
\* ben.c.lambert@gmail.com

**Data Availability Statement:** In a public Zenodo repository, which we cite in the text: https://doi.org/10.5281/zenodo.6771834.

## Abstract

A large range of prognostic models for determining the risk of COVID-19 patient mortality exist, but these typically restrict the set of biomarkers considered to measurements available at patient admission. Additionally, many of these models are trained and tested on patient cohorts from a single hospital, raising questions about the generalisability of results. We used a Bayesian Markov model to analyse time series data of biomarker measurements taken throughout the duration of a COVID-19 patient's hospitalisation for $n = 1540$ patients from two hospitals in New York: State University of New York (SUNY) Downstate Health Sciences University and Maimonides Medical Center. Our main focus was to quantify the mortality risk associated with both static (e.g. demographic and patient history variables) and dynamic factors (e.g. changes in biomarkers) throughout hospitalisation, by so doing, to explain the observed patterns of mortality. By using our model to make predictions across the hospitals, we assessed how predictive factors generalised between the two cohorts. The individual dynamics of the measurements and their associated mortality risk were remarkably consistent across the hospitals. The model accuracy in predicting patient outcome (death or discharge) was 72.3% (predicting SUNY; posterior median accuracy) and 71.3% (predicting Maimonides) respectively. Model sensitivity was higher for detecting patients who would go on to be discharged (78.7%) versus those who died (61.8%). Our results indicate the utility of including dynamic clinical measurements when assessing patient mortality risk but also highlight the difficulty of identifying high risk patients.

**Funding:** The author(s) received no specific funding for this work.

**Competing interests:** The authors have declared that no competing interests exist.

## Introduction

As the coronavirus disease 2019 (COVID-19) pandemic continues to overwhelm many health services, accurate prognosis remains essential to improved clinical care and decisions regarding the equitable allocation of insufficient intensive care resources [1]. Since the beginning of the pandemic, many novel prognostic factors have been identified and applied in prognostic models to predict the course of infection of hospitalised COVID-19 patients [2]. Substantial inequality in the burden of COVID-19 exists, and many social determinants of the outcome of infection have been identified, such as deprivation [3–7]. External validation of novel COVID-19 prognostic factors across a range of different settings is therefore vital. Multivariable models may partially reconcile differences in samples used for model training, though external validation is essential because overfitting and confounding of unknown, yet important, variables are likely to limit the out-of-sample predictive accuracy [8]. Indeed, a systematic validation of 22 prognostic models to an external dataset found none performed better than using the best univariable predictor: age [9]. In an additional study of 107 surveyed prognostic models, many were found to suffer from small sample sizes and have a high risk of bias in the dataset participants [2].

A number of biomarkers at presentation, including C-reactive protein, lymphocyte count, oxygen saturation and urea concentration, are important predictors of hospitalised COVID-19 patient deterioration (defined as the requirement of ventilatory support, critical care or death) and were included in a recently developed prognostic model of patient deterioration which achieved robust predictive accuracy (C-statistic: 0.77) when internally and externally validated on a dataset of 66,705 patients [10]. Similarly, peripheral oxygen saturation, urea level and C-reactive protein at presentation are used to predict patient mortality [11]. Patients are, however, admitted to hospital at different states of disease progression, and their biomarkers change throughout the course of hospitalisation [12–15]. Emerging evidence indicates a number of time-dependent biomarkers changes may therefore be useful prognostic factors: increases in platelets and eosinophil percentage are indicative of reduced mortality risk, whilst increases in alkaline phosphatase may indicate increased mortality risk [15]. Incorporating dynamic changes in biomarkers can improve the predictive accuracy of prognostic models when internally validated [14, 15], but the external validation of these prognostic factors is still required. The role of time-dependent biomarkers in different patients remains a key question [16]. We previously developed a prognostic Markov model, which allows the quantification of daily mortality risk and the impact of dynamic changes in biomarkers on this quantity and fit the model to data from State University of New York (SUNY) Downstate Medical Center [15]. In this study, we fit the model to new data from a different New York hospital: Maimonides Medical Center (henceforth "Maimonides"). We then compare the impact of dynamic changes in patient biomarkers on in-hospital mortality risk (i.e. patient outcomes), across SUNY and Maimonides. In doing so, we obtain an external validation of the model. More importantly, this allows us to appraise the use of dynamic biomarker measurements for determining patient mortality risk in hospitalised COVID-19 patients, which is our main contribution.

## Materials and methods

### Case selection, data extraction and processing

Study approval was obtained from the State University of New York (SUNY) Downstate Health Sciences University Institutional Review Board (IRB#1595271–1) and Maimonides Medical Center Institutional Review Board/Research Committee (IRB#2020–05-07).

A retrospective query was performed among the patients who were admitted to SUNY Downstate Medical Center and Maimonides Medical Center with COVID-19-related symptoms, which was subsequently confirmed by RT PCR, from the beginning of February 2020 until the end of May 2020. Stratified randomization was used to select at least 500 patients who were discharged and 500 patients who died due to the complications of COVID-19. Patient outcome was recorded as a binary choice of "discharged" versus "COVID-19 related mortality". Patients whose outcome was unknown were excluded. Demographic, clinical history and laboratory data were extracted from the hospital's electronic health records. The raw data were cleaned and processed for analysis as described in §S1.2. We make the data for this study available through a Zenodo repository [17].

## Estimating risk of mortality for variables available at presentation

To compare the factors affecting mortality risk across the two hospitals, we calculated the odds ratios (ORs) for each of the variables available at presentation. To do so, we converted the initial biomarker values to binary categories: above (1) or below (0) the pooled sample mean across the two hospitals (in our Markov model, discussed later in Methods, we allow continuous, opposed to binarised impacts of variables on patient outcomes). Laboratory test values at presentation were included only if 150 or more patients in each of the hospitals had data for this test available. If laboratory tests were repeated on the first day of admission, we took the mean value taken on this day to be the value at presentation. Odds ratios were calculated for each variable by estimating the proportion dying for each subgroup and then taking the ratio of these proportions. We assumed the observed counts of individuals expiring were binomially distributed,

$$X_0^j \sim \mathcal{B}(N_0^j, \theta_0^j), \quad X_1^j \sim \mathcal{B}(N_1^j, \theta_1^j), \tag{1}$$

where $j$ indicates the binary variable under consideration (for example, whether an individual was aged 0–40 or whether they had a history of asthma); $X_0^j$ and $X_1^j$ indicate the counts of individuals dying for the two subgroups (e.g. whether an individual was aged 0–40 or not); $N_0^j$ and $N_1^j$ are the observed counts of individuals in the two subgroups; and $\theta_0^j$ and $\theta_1^j$ are the estimated proportions dying in the corresponding subgroups. Parameters were estimated using a Bayesian framework: the estimated ratio of $\theta_1^j/\theta_0^j$ defined the OR for variable $j$ and was estimated by taking 100,000 independent draws from the posterior distributions of each of $\theta_1^j$ and $\theta_0^j$ assuming uniform priors.

## Laboratory value time trends

To determine average trends in laboratory values over the course of a patient's hospitalisation, we carried out a series of regressions for each laboratory test stratified by patient outcome and hospital. To do so, we calculated the percentage change in each patients' biomarker values relative to their values at presentation. These were scaled to have a mean of 0 and standard deviation of 1. Infinite (i.e. when the first biomarker value was zero), missing or extreme observations (the absolute value of the percentage change exceeded the 98% quantile) were excluded from the regression so that our results focused on the bulk of observations, opposed to the extremes. The percentage change in test value was modelled as a function of a quadratic time trend, allowing for fixed effect trends but including individual patient slopes of both the linear and quadratic terms of the trend. These models were estimated in a frequentist framework using the lme4 R package [18], and we extracted the fixed effect estimates of the trends for each model.

## Bayesian Markov models of dynamic risk

The univariate OR estimates described thus far do not account for the impact of other covariates when determining risk. Furthermore, these methods consider a static outcome (whether a patient dies at some point during their hospitalisation) and do not account for the time taken for the outcome to occur, or allow dynamic variables to be included. A patient's underlying risk of death may, however, change throughout the course of their hospitalisation, which can be indicated by changes in certain biomarkers [15].

Here, we briefly describe a multivariate discrete time Markov model which aims to identify the importance of different prognostic factors on COVID-19 mortality risk and estimate the change in individual patients' mortality risk throughout the course of hospitalisation. (The model has previously been described more fully here: [15].) The model specifically accounts for the competing risks of discharge and death. Note, it is possible to use cause-specific hazards models (Cox regressions for each event of interest, treating the other event as censored) to estimate the cumulative incidence function but, in these models, it is not possible to assess the impact of individual covariates on the cumulative incidence function [19, 20]. By considering the sequence of outcomes for each day each patient was in hospital, Markov models can simultaneously account for the risk of discharge versus mortality: on the first day, patients are admitted and begin in the "hospital" state; at the end of the first day, they either remain in hospital or transition to the "discharged" or "death" states. On subsequent days, patients that remained in hospital can undergo the same possible transitions. The probabilities different transitions occur were modelled as a function of each patient's demographic characteristics, comorbidities, laboratory test values at presentation and dynamic trends in laboratory test values (as measured by their percentage changes relative to their values at presentation). A schematic of the model is provided in Fig 1. The un-normalised probabilities of each possible transition are modelled using a log link:

$$q_{it}^{\text{discharged}} = \exp(\alpha_{0i} + \boldsymbol{\alpha}_1' \boldsymbol{x}_{it}), \quad q_{it}^{\text{death}} = \exp(\beta_{0i} + \boldsymbol{\beta}_1' \boldsymbol{x}_{it}), \quad q_{it}^{\text{hospital}} = 1, \qquad (2)$$

where $i$ indicates a given patient; $t$ indicates the day of hospital stay post-admission for a given patient; $\boldsymbol{\alpha}_1$ and $\boldsymbol{\beta}_1$ are vectors of regression coefficients relating to the vector of (potentially time-varying) regressors in $\boldsymbol{x}_{it}$; $\alpha_{0i}$ and $\beta_{0i}$ are patient-specific intercepts. The normalised probabilities of transitions between the states are then given by the ratio of the un-normalised

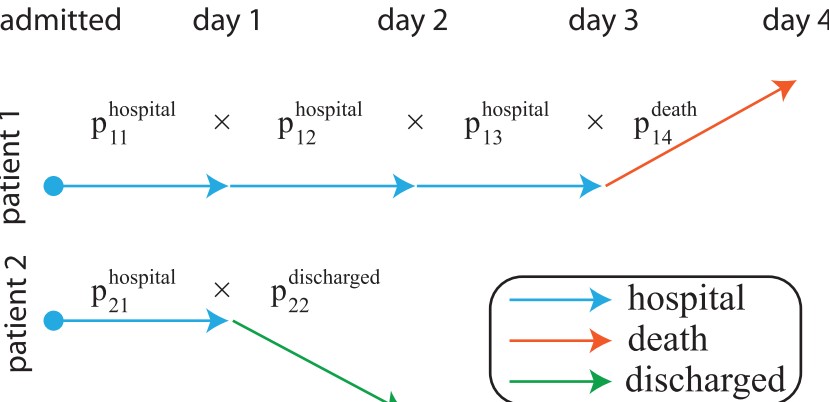

**Fig 1. Markov model of patient trajectories during clinical care.** For two hypothetical patients, we illustrate how the probability of their observed trajectory is calculated. Note that $p_{it}$ refers to the probability of an observed transition, which is a function of the patient ($i$) and day ($t$): the time-dependence of probabilities is realised through Eq (2) and is due to (potential) changes in covariates.

probabilities to the sum of all these: $q_{it}^{\text{total}} = q_{it}^{\text{discharged}} + q_{it}^{\text{death}} + q_{it}^{\text{hospital}}$; so that, for example, $p_{it}^{\text{discharged}} = q_{it}^{\text{discharged}} / q_{it}^{\text{total}}$.

Using this model, we performed six separate regressions, each with different groups of independent variables (i.e. different $\boldsymbol{x}_{it}$ in Eq (2)). In the first of these, we included only a single variable in an analysis to examine the influence of each variable in isolation. The second regression included patient demographic characteristics ("patient" variables), including their age, sex and ethnicity, and the day they were admitted to hospital. The third regression ("pat. + comorbidities") supplemented the patient variables with the recorded comorbidities for each patient: whether they had hypertension, diabetes etc. ($n$ = 13 conditions in total). The fourth regression ("admission") supplemented the third with the initial measurements for each patient for each of the $n$ = 18 clinical tests common across the two hospitals (one test, MCHC, was dropped from these regressions since it is directly calculated from MCH and MCV). The fifth regression ("post-admission") then included the percentage changes in each clinical test measurement from the initial values for each patient. Both the initial values and the dynamic values were scaled to give a mean of 0 and a standard deviation of 1, so that the ORs were estimated on a scale that was consistent across the different laboratory tests and represented the typical clinical variation in these values. The biomarkers included in the study along with the acronyms used are given in S1 Table in S1 File. The final regression considered used both the static variables of the pat. + comorbidities regression and the raw values of the tests (standardised by subtracting the sample mean and dividing through by the sample standard deviation). The aim of this regression was to investigate whether relative changes from baseline or, rather, the absolute covariate values which were most predictive of outcomes.

The model was estimated in a Bayesian framework using the Stan's NUTS sampler [21, 22]. We used priors for the regression coefficients that induce sparsity: meaning that only the most predictive covariates would be estimated to have non-zero effects. The priors for the parameters are shown in S2 Table in S1 File. The univariate models were run for 2000 iterations; the multivariate models were run for either 2000 iterations, then a further 2000–6000 iterations if not converged. In all cases, we ran the model using four chains with the first half of iterations discarded as warm-up. The Markov chains satisfied $\hat{R} < 1.01$ and bulk- and tail-ESS > 400 for all parameters, consistent with convergence. The Stan code for the model is provided in §S1.3.

## Generalisation of predictions

Next, we assessed whether the sets of factors considered in this paper can be used to predict patient outcomes that generalise across both hospitals. To do so, we fitted the Markov model to data from each hospital in turn, then used it to predict patient outcomes (i.e. whether the patient ultimately died in hospital or was discharged) in the other held-out hospital. In this analysis, we did not consider the time taken for death or discharge to occur, and future work could consider also these outcomes (although our previous work has demonstrated that predicting timings is likely difficult [15]). As discussed in Methods, we scaled both the initial laboratory values and the dynamic values using the sample mean and standard deviation. When performing between-hospital prediction, we used the mean and standard deviation of values of the training hospital to scale variables in the independent hospital test set. This ensured that we only used information available in the training set when making predictions.

To check that the Markov model provided a reasonable fit of the underlying data, we performed a series of posterior predictive checks (PPCs) (see, for example, [23, 24]). But, in order to assess their generalisation of the fitted models, we performed the PPCs on independent hold-out sets. For the Markov model with the post-admission set of variables, we compared the model-estimated and actual mortality rates, separately for models trained on data from

SUNY and Maimonides. In S1 Fig in S1 File, we show the estimated (black point-ranges) and estimated (orange points) mortalities across groupings of our binary predictor variables for a model fit to data from SUNY and used to predict the outcome in an independent test set also from SUNY. These graphs indicate a good correspondence in the majority of cases. In S2 Fig in S1 File, we show a similar plot but for the dynamic biomarkers where we compare mortality rates for groups of individuals with last recorded laboratory values above or below the mean: again, this plot illustrates a reasonable fit. In S3 & S4 Figs in S1 File, we show the same plots but when fitting to data from SUNY but predicting outcomes in Maimonides. These fits were noticeably poorer than for the within-SUNY fits, although the general trends in outcome across the binarised groups tended to be similar. In S5-S8 Figs in S1 File, we repeat the same analysis, but using Maimonides as the data used to train the model.

## Results

### There were notable differences in demographics across the hospitals

The hospital cohorts ($n = 553$ patients in SUNY; $n = 987$ in Maimonides) differed in demographic variables and underlying comorbidities (Table 1). Patients of SUNY predominantly

**Table 1. Summary characteristics of patient groups from the two hospitals.** Note, that in some cases, data were missing meaning that patients counts across all shown categories do not aggregate to $n = 553$ for SUNY and $n = 987$ for Maimonides.

| Variable | SUNY | Maimonides |
|---|---|---|
| outcome: discharged | 342 (61.8%) | 496 (50.3%) |
| outcome: expired | 211 (38.2%) | 491 (49.7%) |
| sex: female | 271 (50.3%) | 437 (44.3%) |
| sex: male | 268 (49.7%) | 550 (55.7%) |
| ethnicity: black | 472 (86.8%) | 119 (12.3%) |
| ethnicity: hispanic | 17 (3.1%) | 2 (0.2%) |
| ethnicity: other or unrecorded | 39 (7.1%) | 271 (27.5%) |
| ethnicity: white | 25 (4.6%) | 595 (61.7%) |
| age: 0–40 | 26 (4.8%) | 116 (11.8%) |
| age: 40–50 | 43 (7.9%) | 57 (5.8%) |
| age: 50–60 | 93 (17.2%) | 112 (11.3%) |
| age: 60–70 | 140 (25.8%) | 201 (20.4%) |
| age: 70–80 | 137 (25.3%) | 210 (21.3%) |
| age: 80+ | 103 (19.0%) | 291 (29.5%) |
| asthma | 24 (4.4%) | 89 (9.0%) |
| cancer | 16 (2.9%) | 89 (9.0%) |
| cerebrovascular disease | 25 (4.6%) | 80 (8.1%) |
| congestive heart failure | 23 (4.2%) | 292 (29.6%) |
| chronic kidney disease | 19 (3.5%) | 76 (7.7%) |
| copd | 25 (4.6%) | 96 (9.7%) |
| coronary artery disease | 44 (8.0%) | 391 (39.6%) |
| dementia | 13 (2.4%) | 120 (12.2%) |
| diabetes | 229 (41.9%) | 348 (35.3%) |
| endstage renal disease | 54 (9.9%) | 50 (5.1%) |
| hepatitis | 4 (0.7%) | 24 (2.4%) |
| hyperlipidemia | 103 (18.8%) | 270 (27.4%) |
| hypertension | 350 (64.0%) | 516 (52.3%) |

self-reported as black, whereas those of Maimonides predominantly self-reported as white. Diabetes was more prevalent in the SUNY cohort, whereas coronary artery disease and congestive heart failure were more prevalent in the Maimonides cohort. The presence of multiple conditions within individual patients differed substantially between the two cohorts (S9 Fig in S1 File). There were minor differences in the distributions of the laboratory test values at admission (available across both hospitals) of the two hospital cohorts, with the exception of BASO PCT and MCHC (S10 Fig in S1 File).

### There was consistent mortality risk associated with static factors

To compare the mortality risk for those variables available at presentation (including demographic variables, comorbidities and laboratory test values at admission), which were common across the two hospitals, we estimated the ORs measuring the risk of death associated with each of the variables.

There was a significant positive correlation between the OR estimates of the demographic and comorbidity variables of the two hospitals (using median posterior estimates: $\rho = 0.82$, $t_{19} = 6.33$, $p < 0.01$; here and throughout, $\rho$ indicates Pearson correlation coefficient estimates) (Fig 2A). In Maimonides, the variables tended to be less associated with risk than for SUNY (regression slope of posterior median estimates for Maimonides on those from SUNY: $\hat{\beta} = 0.65$, $t_{19} = 3.48$, $p < 0.01$ against $H_0$: $\beta = 1$). Similarly, there was a significant positive correlation between the OR estimates of the laboratory values at admission between the two

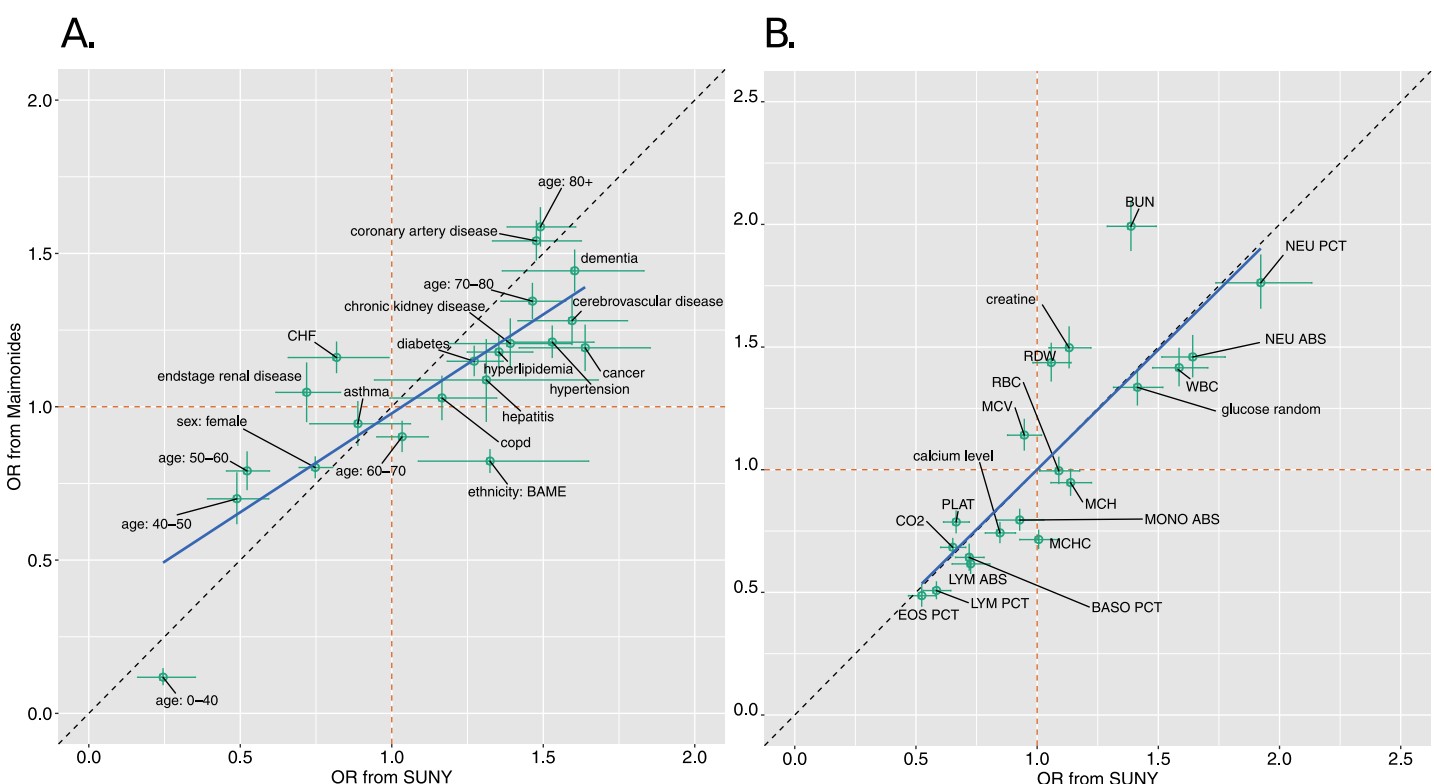

**Fig 2. Comparing univariate mortality ORs across the hospitals.** The two panels compare ORs associated with demographic and disease history variables (panel A) and initial laboratory test values (panel B). Points show the posterior median ORs; the whiskers display the 25% and 75% posterior quantiles. The orange dashed lines show the OR = 1 cases; the dashed black lines indicate equality in the ORs across the two hospitals. The blue line shows least squares regression lines using the posterior median ORs.

hospitals ($\rho$ = 0.85, $t_{17}$ = 6.77, $p < 0.01$), and the regression slope was not significantly different from 1 ($t_{17}$ = 0.22, $p > 0.05$) indicating there was no systematic differences in ORs between hospitals for these variables (Fig 2B).

## The dynamics of biomarker values were remarkably similar across the hospitals for patients with the same outcomes

We next considered dynamic changes in the $n$ = 19 laboratory biomarkers which were available across both hospitals, which we plot in Fig 3. This illustrates that, irrespective of patient outcome, there is considerable inter-patient variability in the time series of these biomarkers.

To compare the average dynamics of laboratory test values for patients throughout the course of their hospitalisation, we also estimated hospital-specific time trends for each patient

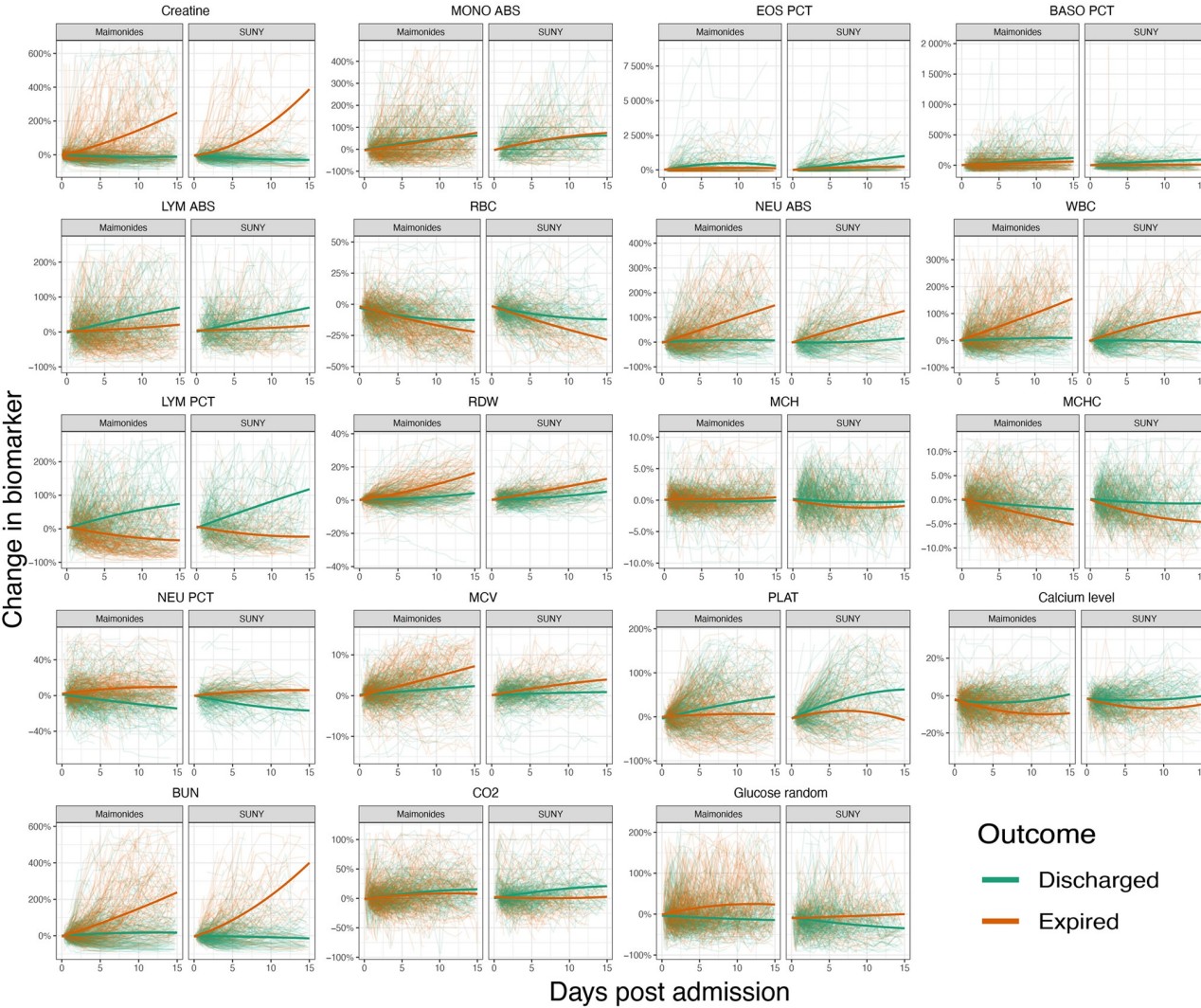

**Fig 3. Comparing time trends in laboratory values across hospitals.** The horizontal axis shows the days post admission and the vertical axis shows the percentage change in biomarker values from their initial values. Each panel displays trends for an individual biomarker; within the subpanels of each of these, we show the results for each of the hospitals. Individual graphs show the dynamics of the individual patients' laboratory values (thin coloured lines) and the time trends (thick coloured lines) estimated assuming a quadratic regression function. Line colouring indicates the outcome of an individual patient (thin lines) or overall group being considered for regressions (thick lines). Note that for plotting we only display data up until day 15 post admission, since, after this point, there were relatively few patients still hospitalised.

group (see Methods). Across the majority of variables, there was a high degree of correspondence in these average trends across the two hospitals (Fig 3). Indeed, the correlation between the regression estimates of the percentage change in laboratory values at 15 days post admission (after this point, only a minority of patients were still hospitalised) was correlated across the hospitals: for both the discharged and expired groups, these correlations were significant and positive (discharged: $\rho = 0.94$, $t_{17} = 11.25$, $p < 0.01$; expired: $\rho = 0.94$, $t_{17} = 11.58$, $p < 0.01$).

## The biomarkers associated with mortality risk were generally similar across the hospitals

In Fig 4, we compare the ORs associated with daily mortality risk for each of the common biomarkers across the two hospitals as derived from the Markov model (described in Methods). Across the univariate and multivariate model estimates, there was strong positive correlation in the ORs between the hospitals (univariate: $\rho = 0.84$, $t_{16} = 6.52$, $p < 0.01$; multivariate: $\rho = 0.65$, $t_{16} = 3.45$, $p < 0.01$; in both cases, using posterior median estimates). With few exceptions the estimates agreed in terms of their "sign": for the univariate model, 15/18 tests had posterior median estimates where either both odds ratios were above one across the two hospitals or both were below one; for the multivariate model, the corresponding figure was 14/18 tests. A notable outlier was MCV, which was estimated to have a substantially stronger effect in the Maimonides cohort in the multivariate model, although this was not recapitulated in the univariate analysis suggesting caution interpreting this further.

Based on these estimates, increases in MCV, decreases in LYM PCT, and decreases in $CO_2$ throughout a patient's stay were associated with the strongest increase in mortality risk.

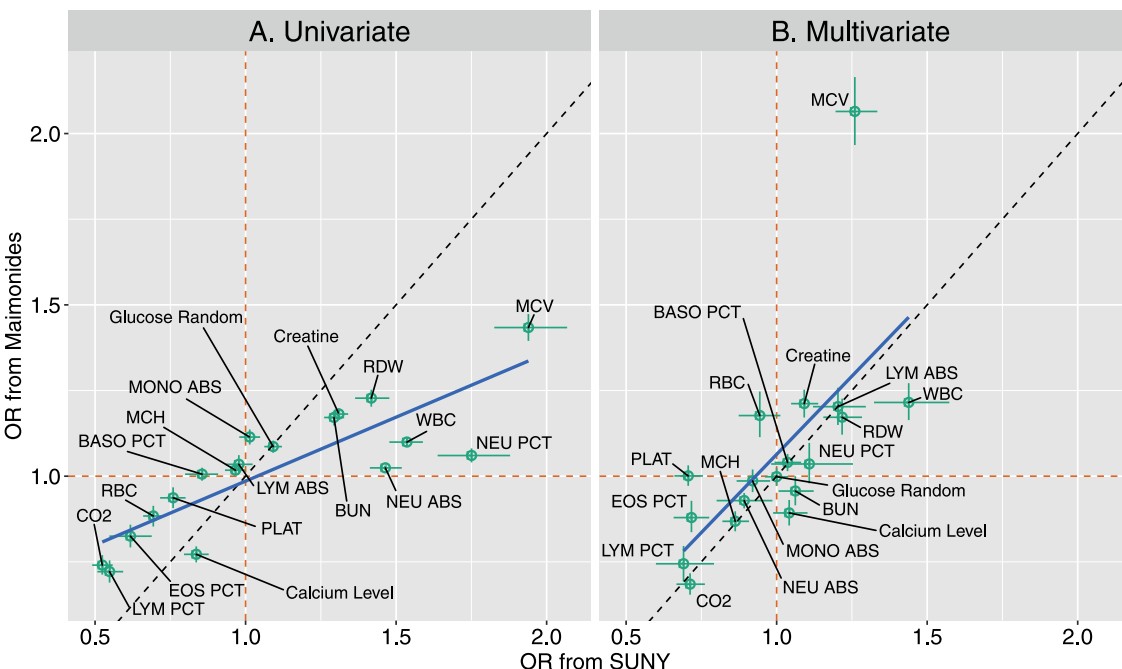

**Fig 4. Comparing ORs for dynamic laboratory measurements.** Panels A/B compare the ORs for the univariate/multivariate models. The horizontal axes displays the ORs for daily mortality risk from SUNY and the vertical axes show the ORs from Maimonides. Points show the posterior median ORs; the whiskers display the 25% and 75% posterior quantiles. The orange dashed lines show the OR = 1 cases; the dashed black lines indicate equality across ORs calculated across the hospitals. The blue line shows least squares regression lines using the posterior median ORs.

### The models generalised well across hospitals and fared better in predicting the outcome of patients who were discharged

Across the different regressor sets, out-of-sample predictive accuracy was consistent across the hospitals (Fig 5A). In all cases, the posterior median predictive accuracy using data from Maimonides resulted in slightly higher prediction accuracy than when using data from SUNY: likely due to the higher sample size for Maimonides. The results also show the predictive power of dynamic laboratory measurements (included in the "post-admission" set), which resulted in a substantial boost in accuracy across both hospitals over a model including only those available at admission ("admission").

We next used the model using all available post-admission variables to probe its predictive performance for those groups of patients who went on to be discharged and died. To do so, we pooled predictions across both independent hospital testing sets: note, that in both cases, these predictions were formed using out-of-sample testing sets. The resultant *confusion matrix* is shown in Fig 5B. This indicates that the model had a higher sensitivity to determine patients that would eventually be discharged (posterior mean: 79.2%) compared to those who would go on to die (61.0%).

We next assessed the reduction in accuracy when predicting patient outcomes in the same hospital versus a different hospital. We did this by using validation sets either comprised of separate data from the same hospital ("within') or a different hospital ("between"). In S11 Fig in S1 File, we show the predictive accuracy for models fitted using data from SUNY (left panel) and Maimonides (right panel). Point colour indicates whether the validation set comprised patients from within the same hospital (green) or a different hospital (orange). Note that, in this analysis, the requirement for an independent within-hospital datasets to fit the model

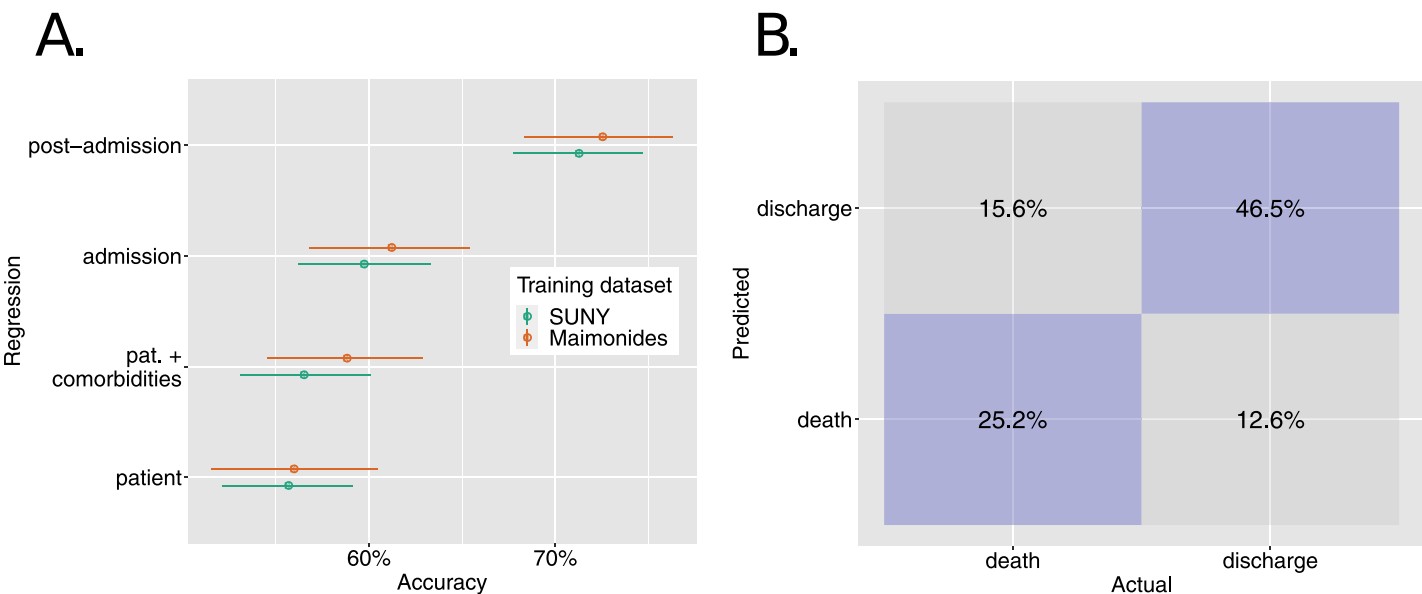

**Fig 5. Model predictive accuracy in patient outcomes for held-out hospital.** Panel A shows the accuracy in predicting outcomes across four regression sets. Colours indicate the hospital whose data was used to train the model: so, for example, "SUNY" indicates that data from this hospital was used to fit the model which was then tested on data from Maimonides. The horizontal axis shows the accuracy in predicting patient outcomes (i.e. death or discharge) using a Markov regression model with covariate sets as named on the vertical axis. The points and whiskers indicate the posterior medians and 2.5%-97.5% posterior intervals for the percentage of patients whose outcome was correctly determined across posterior draws. Panel B shows a confusion matrix for between-hospital prediction using Markov model with the post-admission covariate set. Here, the values show the mean percentage of each outcome type correctly predicted across all posterior samples.

meant that the training datasets were smaller than those used to produce Fig 5A: resulting in slightly lower overall accuracy. In almost all cases, median predicted accuracy when predicting outcomes within the same hospital was higher than that when predicting those in a different hospital. The difference, however, was relatively small (mean difference in posterior medians: 2.1%), indicating that the predictions generalised well from one location to another.

To determine whether there were subgroups of patients where the model performed better or worse, we examined the factors that influenced the predictive accuracy for those patients who went on to die in each of the hospitals. To do so, we used the models that were trained on all the data from one hospital (i.e. those used to produce Fig 5A). We then used a random forest to predict posterior median predictive accuracy for each of the patients, as a function of their time-invariant characteristics. Then using the "impurity" measure of variable importance, we identified those variables that were associated with differences in predictive accuracy (S12 Fig in S1 File). The top three factors were: the time since the first patient was admitted to that hospital with COVID-19, whether the patient had a history of coronary artery disease and whether their self-reported ethnicity was categorised as BAME. These three variables were then included in a linear regression to predict median predictive accuracy. This regression indicated that having a history of coronary artery disease led to improved predictive power ($\beta = 0.14$, $t_{478} = 5.66$, $p < 0.01$); other factors were insignificant.

### Relative changes in biomarker values and their raw values are similarly predictive of outcomes

We tested the hypothesis that the biomarker values themselves opposed to their relative changes from a patient-specific baseline were most important in determining outcomes by fitting our multivariate Markov model using the biomarker values as covariates: we term this model the "absolute values" analysis to distinguish it from the "relative changes" analysis.

To do so, we compared the predictive power of the absolute values analysis with that of the relative changes one. In S13 Fig in S1 File, we show the predictive accuracy in determining patients' outcomes for an independent hold-out hospital across the two analyses. This illustrates very similar accuracies across the two analyses, and that it is not possible for us to conclude whether the absolute biomarker values or the changes from baseline are most clinically relevant.

To explore whether the two analyses led to different conclusions about clinically relevant changes in biomarker values, we compared the ORs for mortality risk across the analyses. The ORs from the absolute values analysis were, like those from the relative changes analysis, very consistent across the hospitals (S14 Fig in S1 File; $\rho = 0.88$, $t_{16} = 7.40$, $p < 0.01$; using posterior median estimates). We also compared the ORs across the two analyses (S15 Fig in S1 File). The ORs produced across the two analyses were of the same sign for 15/18 biomarkers for SUNY and for 15/18 for Maimonides, and the ORs were strongly positively correlated for both hospitals (SUNY: $\rho = 0.77$, $t_{16} = 4.88$, $p < 0.01$; Maimonides: $\rho = 0.94$, $t_{16} = 11.4$, $p < 0.01$; both using posterior median estimates). Overall, there were six biomarkers whose increases were associated with increases in mortality risk across the two analyses and hospitals: BASO PCT, Creatine, LYM ABS, MCV, RDW and WBC; and a set of seven biomarkers whose increases led to reductions in risk: $CO_2$, EOS PCT, Glucose Random, LYM PCT, MCH, MONO ABS and NEU ABS.

### Discussion

A number of studies have demonstrated that dynamic changes in certain laboratory tests may have potential as COVID-19 prognostic factors [12, 14, 15]. Here, we demonstrate the

external validation of a number of these dynamic biomarkers. In accordance with existing studies, we find a number of biomarkers at presentation (or when measured at a single time-point) increased mortality risk across both hospital cohorts (univariate ORs): these included eosinopenia [25], thrombocytopenia [26, 27], lymphocytopenia [27, 28] and increased blood urea concentration (in our case indicated by BUN) [11]. In addition, we quantified the reduction in mortality risk associated with dynamic variation in biomarkers and, across the two hospitals found remarkably consistent estimates. Interestingly, we identified biomarkers that have little prognostic value at presentation whereas their dynamic changes do: increases in MCV, for example, increased mortality risk. These results highlight the potential importance of measuring dynamic changes in biomarkers for patient prognosis. Our model could better predict outcomes for patients who went on to be discharged opposed to those who eventually died, indicating the challenges in assessing mortality risk in hospitalised COVID-19 patients.

Our study suffered from a number of limitations. Data availability limited the prognostic factors tested, and future work is therefore required to quantify the mortality risk associated with dynamic changes in other prognostic factors that are known to be important at presentation. These include abnormal biomarkers of inflammation, myocardial injury, acute respiratory distress syndrome (ARDS) and coagulopathy [26, 28]. We also did not include time-dependent changes in certain chemokines and cytokines, which can also indicate disease progression [12, 29]. Additionally, we did not account for the potential impact of patient treatment on dynamic changes in biomarkers or on outcomes. Mechanical ventilation of patients with ARDS, for example, is used to maintain certain arterial $pCO_2$ values, and both mechanical ventilation and certain COVID-19 pharmaceutical treatments can influence inflammatory markers [30, 31]. We considered patients solely hospitalised during early to mid-2020 within a single region (New York), but novel variants and existing immunity may alter survival [32]. Within certain settings, patient survival has improved throughout the course of the pandemic [33], and temporal recalibration of multivariate regression models, which aim to quantify the OR of survival for different prognostic factors, is therefore necessary to ensure survival is not under- or overestimated [34]. By using a relatively simple model (which assumed a linear functional form on the log-odds scale), we focused on the ability of our model to explain not predict [35]. Using this approach, it is possible that we missed important contributions from the interactions between factors, and future work could investigate the use of models such as Bayesian Additive Regression Trees [36], which allow non-linear interactions between regressors.

Whilst our model performed well across the two cohorts examined, we caution against its use as a dynamic prognostic model in clinical settings. In order for it to be used as thus, any such model requires training and evaluation over a much larger sample size across multiple settings including the full set of factors implicated with risk. The ease of use in a clinical setting and effects on clinicians' behaviour, comparison with existing prognostic models, cost-effectiveness and impact on patient health must also be assessed prior to the implementation of any prognostic model [37].

## Supporting information

**S1 File. Supplementary information: Contains biomarker abbreviations, the priors used, the various posterior predictive checks, graphical comparisons of raw data and assessments of model predictive power.**
(PDF)

## Acknowledgments

IJS wishes to acknowledge funding from the medical research council, grant number: MR/R015600/1. The authors also wish to thank the study participants for agreeing to the use of their data.

## Author Contributions

**Conceptualization:** Ben Lambert, Isaac J. Stopard, Amir Momeni-Boroujeni, Rachelle Mendoza, Alejandro Zuretti.

**Data curation:** Amir Momeni-Boroujeni, Rachelle Mendoza, Alejandro Zuretti.

**Formal analysis:** Ben Lambert, Isaac J. Stopard.

**Investigation:** Ben Lambert, Isaac J. Stopard, Amir Momeni-Boroujeni.

**Methodology:** Ben Lambert, Isaac J. Stopard.

**Project administration:** Ben Lambert, Alejandro Zuretti.

**Supervision:** Ben Lambert.

**Validation:** Ben Lambert.

**Visualization:** Ben Lambert, Isaac J. Stopard.

**Writing – original draft:** Ben Lambert, Isaac J. Stopard.

**Writing – review & editing:** Ben Lambert, Isaac J. Stopard, Amir Momeni-Boroujeni, Rachelle Mendoza, Alejandro Zuretti.

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
