## [Decision Letter · Decision Letter 0]

16 May 2022

PONE-D-21-35787Using patient biomarker time series to determine mortality risk in hospitalised COVID-19 patients: a comparative analysis across two New York hospitalsPLOS ONE

Dear Dr. Lambert,

Thank you for submitting your manuscript to PLOS ONE. After careful consideration, we feel that it has merit but does not fully meet PLOS ONE’s publication criteria as it currently stands. Therefore, we invite you to submit a revised version of the manuscript that addresses the points raised during the review process.

Reviewers largely found the manuscript well-written and methodologically sound, though they identified some issues of clarity.  A revision of the text is likely to be able to address these issues.

We look forward to receiving your revised manuscript.

Kind regards,

Bryan C Daniels

Academic Editor

PLOS ONE

**Journal requirements:**

**Reviewers' comments:**

Reviewer's Responses to Questions

**Comments to the Author**

1. Is the manuscript technically sound, and do the data support the conclusions?

Reviewer #1: Yes

Reviewer #2: Yes

2. Has the statistical analysis been performed appropriately and rigorously? 

Reviewer #1: Yes

Reviewer #2: Yes

3. Have the authors made all data underlying the findings in their manuscript fully available?

Reviewer #1: Yes

Reviewer #2: No

4. Is the manuscript presented in an intelligible fashion and written in standard English?

Reviewer #1: Yes

Reviewer #2: Yes

5. Review Comments to the Author

Reviewer #1: Review

This manuscript evaluates a previously published Bayesian Markov model of hospitalized COVID-19 patient outcomes on data from 2 New York hospitals early in the pandemic. It is a reasonable approach to model evaluation in this context, which is nice to see, although I have a few concerns and this represents no more than an incremental contribution to the field.

Major Concerns

1. In the post-admission data, is it the change in biomarker values from baseline that is informative or just having data from a later time point (which I would think is generally more predictive of a hospital outcome)? Comparing to a model that uses follow up covariate values would get at this.

2. How would a naive black box machine learning algorithm fare at this prediction problem? This seems like a natural point of comparison, since these models (e.g., random forest, boosting, etc.) are so easy to use and widely available. To make a contribution, any prediction model must distinguish itself in some way from these readily available alternatives.

3. I admit I am relatively ignorant of fitting models to MCHC, MCV and MCH. However, my understanding is that MCHC = MCH / MCV. Consequently, I consider it a little unusual to include all 3 as covariates in a model. Please explain. Could this be the reason that the MCHC and MCV estimates in lines 227-236 (and Fig S7A) seem so off?

Minor Concerns

1. Line 239: It might be worth making explicit mention in this paragraph that the evaluation uses out-of-sample predictions. That is a good approach, and telling the reader that here, up front, may alleviate some fears regarding suitability of the evaluation procedure that would not otherwise be allayed until later.

2. Line 247: Please correct the grammatical error in “groups patients”.

3. Line 248-251: I believe this confusion matrix uses in-sample predictions. Some type of out-of-sample estimator would be more compelling, e.g., cross-validation.

4. Line 270-275: It would be nice to show the entire set of variable importance scores in the supplemental material.

5. Supplemental Figure 7: The caption refers to a “dashed black line”, when the line is solid.

6. Supplemental Figure 7: Panel b doesn’t just zoom in, but refits the blue line on a reduced set of features (which the text notes, but the caption is misleading)

Reviewer #2: This article follows up on an earlier study by the authors that used time-varying covariates to model mortality risk in COVID-19 patients hospitalized in a New York hospital.

The focus of the article is on results obtained using data from an additional hospital, as well using the data from each of the two hospitals for the purposes of external validation of the model trained using data from the other. These are both useful contributions. As such, the article is well-written and clearly articulates its goals, findings, and limitations. The methodology appears to be sound.

However, I do feel that a fundamentally important quantity has not been clearly defined, either in this or the original article. Specifically, what exactly is the 'prediction' made by the dynamic model?

Fig 5A compares the internal and external accuracies of the 'prediction' made by four models. Here, the dynamic 'post-admission' model shows a clear improvement (at around 72% accuracy) over the next best 'admission' model (at around 60%). The prediction made by the admission model is unambiguous: for any particular patient, it is based on the patient covariates at the time of admission.

However, for the dynamic model, a prediction can be made for each patient for every day that the patient stays in the hospital. These will depend on the parameters of the trained model as well as the current values of the patient's biomarkers. Which of these several possible predictions have been used to arrive at the prediction accuracy for the post-admission model in Fig 5A? At a minimum, I think this needs to be clearly stated, especially as the unavailability of the raw data makes it impossible to reproduce the analysis.

If for example, the prediction used is the one on the last day of stay (before death or discharge), then the improvement in accuracy may not be as impressive. In that case, a more appropriate quantity to compare could be the predictions from the admission model using the corresponding day's biomarker values (instead of the values at admission).

Also, regardless of the definition used, I think it may be useful to quantify and visualize the progression of the daily prediction accuracy over hospital stay. The varying length of hospital stay makes this not straightforward, but perhaps the authors could consider a plot of accuracy of prediction made with covariates available k days before actual death or discharge, with k varying from 1 to 10 or so.

I noticed a couple of minor typos around lines 41-42 (grammar) and line 241 (the word 'accuracy' is missing).

6. PLOS authors have the option to publish the peer review history of their article (what does this mean?). If published, this will include your full peer review and any attached files.

Reviewer #1: No

Reviewer #2: No

---

## [Author Response · Author response to Decision Letter 0]

12 Jul 2022

These are within our response to the reviewer comments document, which we include as part of this submission.

---

## [Editor Report · Decision Letter 1]

20 Jul 2022

Using patient biomarker time series to determine mortality risk in hospitalised COVID-19 patients: a comparative analysis across two New York hospitals

PONE-D-21-35787R1

Dear Dr. Lambert,

We’re pleased to inform you that your manuscript has been judged scientifically suitable for publication and will be formally accepted for publication once it meets all outstanding technical requirements.

Kind regards,

Bryan C Daniels

Academic Editor

PLOS ONE
---

## [Editor Report · Acceptance letter]

10 Aug 2022

PONE-D-21-35787R1 

Using patient biomarker time series to determine mortality risk in hospitalised COVID-19 patients: a comparative analysis across two New York hospitals 

Dear Dr. Lambert:

I'm pleased to inform you that your manuscript has been deemed suitable for publication in PLOS ONE. Congratulations! Your manuscript is now with our production department. 

Kind regards, 

on behalf of

Dr. Bryan C Daniels 

Academic Editor

PLOS ONE